# SARS-Cov-2 Interactome with Human Ghost Proteome: A Neglected World Encompassing a Wealth of Biological Data

**DOI:** 10.3390/microorganisms8122036

**Published:** 2020-12-19

**Authors:** Tristan Cardon, Isabelle Fournier, Michel Salzet

**Affiliations:** 1Inserm U1192, University Lille, CHU Lille, Laboratory Protéomique Réponse Inflammatoire Spectrométrie de Masse (PRISM), F-59000 Lille, France; 2Institut Universitaire de France, 75000 Paris, France

**Keywords:** SARS-Cov-2, interactomics, ghost protein, drug repurposing, alternative protein, non-coding RNA, alternative ORF, OpenProt, innate immune response, anti-viral response

## Abstract

Conventionally, eukaryotic mRNAs were thought to be monocistronic, leading to the translation of a single protein. However, large-scale proteomics have led to a massive identification of proteins translated from mRNAs of alternative ORF (AltORFs), in addition to the predicted proteins issued from the reference ORF or from ncRNAs. These alternative proteins (AltProts) are not represented in the conventional protein databases and this “ghost proteome” was not considered until recently. Some of these proteins are functional and there is growing evidence that they are involved in central functions in physiological and physiopathological context. Based on our experience with AltProts, we were interested in finding out their interaction with the viral protein coming from the SARS-CoV-2 virus, responsible for the 2020 COVID-19 outbreak. Thus, we have scrutinized the recently published data by Krogan and coworkers (2020) on the SARS-CoV-2 interactome with host cells by affinity purification in co-immunoprecipitation (co-IP) in the perspective of drug repurposing. The initial work revealed the interaction between 332 human cellular reference proteins (RefProts) with the 27 viral proteins. Re-interrogation of this data using 23 viral targets and including AltProts, followed by enrichment of the interaction networks, leads to identify 218 RefProts (in common to initial study), plus 56 AltProts involved in 93 interactions. This demonstrates the necessity to take into account the ghost proteome for discovering new therapeutic targets, and establish new therapeutic strategies. Missing the ghost proteome in the drug metabolism and pharmacokinetic (DMPK) drug development pipeline will certainly be a major limitation to the establishment of efficient therapies.

## 1. Introduction

Because proteins are the end products of gene expression, they have a major impact on cell regulation, thus being main targets for the development of new drugs and therapies. Therefore, holistic approaches must be developed to grasp the proteome in its completeness and find out how it relates to the upstream genes it is issued from. Grasping the proteome can be difficult because of the broad dynamic ranges its spans on (i.e., > 7 orders of magnitude from 1 copy to up to 10 million per cell) when compared to transcriptome (only 3–4 orders of magnitude) [1]. However, thanks to the last generation of liquid chromatography-mass spectrometry (LC-MS) instrumentation, > 5000 proteins can be identified in a single run experiment by large-scale bottom-up proteomics [2]. Both bottom-up and top-down proteomic approaches are very powerful; though they do show a major drawback since the protein identification is based on databank interrogation. Databanks are thus critical to large-scale proteomic approaches, since only proteins referenced in the database can be identified. A large part of the proteins in databases, such as UniProtKB/Swiss-Prot, which is the reference database in proteomics [3], is predicted from genes according to well established rules. Thereof, only >100 codon sequences of mRNA starting with an “AUG” and presenting the favorable consensus Kozak motive are translated into a single protein accordingly to the admitted idea that eukaryotes are monocistronic. The single protein product expected from gene translation is designated as the reference protein (RefProt).

However, eukaryotic translation was finally demonstrated to be polycistronic as already suspected in the late 1990s by M. Kozak [4]. Indeed, alternative translation mechanisms, such as the reinitiation or the leaky-scanning, leading to translation from alternative ORFs (AltORFs), were already described by that time; though those has remained considered as an epiphenomenon. Hence, a huge number of proteins were lacking from protein databases and have simply remained invisible to all proteomic studies, representing, thereby, a ghost proteome. This ghost proteome was eventually unveiled by two distinct approaches, one using ribosome profiling [5], and the second, MS-based proteomics. In ribosome profiling, many possible fixations of ribosome were described from non-coding RNA (ncRNA) and untranslated region (UTR) of mRNA [6,7], highlighting the existence of non-expected protein products in mammalians. From proteomic data, by using novel databases that included protein predictions translated from AltORFs novel protein sequences were identified, filling the gap of good quality data remaining unmatched after conventional database interrogation (>10% data) [8]. These proteins, designed as alternative proteins (AltProts), are neither proteoforms, nor proteins issued from alternative splicing. Some show sequence similarities with proteins carried by other mRNA, but the others present totally new amino acid sequences. Finally, identified AltProts are found to be translated, either from mRNA including from the non-coding 5′ & 3′ UTR or a frame shift (+1 or 2 nucleotides) in the CDS of the RefProt, or from ncRNA [9]. Overall, large-scale bottom-up [9,10,11,12] and top-down [13,14] proteomics have enable the identification of an important number of these AltProts. Very importantly, AltProts were also shown to be functional and carrying important cell functions [12,15,16,17]. In a way, the rediscovery of the “lost world” of protein products will open a new page in the history of biological mechanisms.

A total of ~450,000 proteins has ultimately been predicted in humans and are publicly available through the OpenProt [18] database. This is about 20-fold more than yet estimated from conventional databases (20,353 entries in June 2020 for reviewed RefProt). It is thus possible to gain incredible knowledge by considering AltProts in already generated data. Previously, proteomic data reuse have enabled the discovery of the ghost proteome interactome using cross-linking MS (XL-MS) data from HeLa cells [19,20]. In this study, AltProts were found to be interacting with RefProts involved in protein translation regulation as evidenced by the participation of AltATAD2 in the RPL10/AUF1 complex [20]. Since the study of glioma cell line (NCH82) under activation by a protein kinase A activator, inducing a cellular phenotypic change has confirmed the presence of AltProts in the signaling pathways of protein translation. AltProts were also shown interacting with cytoskeleton proteins (e.g., AltTRNAU1AP, AltMAP2, and AltEPHA5 interacting with TPM4) [10].

Based on our experience with AltProts, we were interested in finding out their involvement in development of the SARS-CoV-2 virus, responsible for the 2020 COVID-19 outbreak. Thus, we scrutinized the recently published data by Gordon and Krogan team [21] on the SARS-CoV-2 interactome with host cells by co-IP in the perspective of drug repurposing. In this work, the team have cloned the viral target proteins with a 2XStrep tag based on the GenBank sequence for SARS-CoV-2 isolate 2019-nCoV/USA-WA1/2020, accession MN985325, downloaded on 24 January, 2020. Tagged protein are express in human cells (HEK-293T/17) in order to identify the physical interaction partners of these proteins. Thus, by affinity purification coupled to mass spectrometry (AP-MS), 332 high confidence interactions were identified between the viral protein and the host. Based on these identifications, gene ontology enrichment and analysis were performed to identified pathway involved on the viral infection; moreover, some structure prediction of the viral proteins was performed with some measurements of interaction, e.g., ORF6 and NUP98-RAE1 complex. Finally, drug repurposing, targeting the identified host proteins, was proposed, based on chemoinformatics analysis of SARS-CoV-2-interacting partners and molecular docking. In this way, 69 FDA approved therapeutic compounds were evaluated against SARS-CoV-2 infection; some have been part of viral growth and cytotoxicity assays. Techniques and methodology are described in detail in the article of April 30, 2020: “A SARS-CoV-2 protein interaction map reveals targets for drug repurposing”.

## 2. Material and Methods

### Ghost Proteins Databases

The study was carried out using OpenProt database (www.openprot.org) [18,22]. This database is derived from the predicted H. Sapiens alternative proteins (GRCh38.p5, Assembly: GCA_000001405.20). This database compiles all proteins coming from non-coding regions of mRNA, such as 5′&3′ UTR, shift in reading frame in +2 or +3, and the proteins discovered coding in ncRNA. Moreover, to this database, the RefProt from UniProtKB is added, for a total of 658,263 entries. Proteome Discoverer 2.3 (PD2.3) with label free quantification node is used to analyze the RAW data from ProteomeXchange consortium via the PRIDE repository dataset, number PXD018117 [21]. The following parameters apply on PD2.3: trypsin as enzyme, 2 missed cleavages, methionine oxidation as variable modification, and carbamidomethylation of cysteines as static modification, precursor mass tolerance: 10 ppm and fragment mass tolerance: 0.6 Da. The validation was performed using Percolator with an FDR set to 0.001%. A consensus workflow was then applied for the statistical arrangement, using the high confidence protein identification and at least one unique peptide for identified proteins.

The identified proteins are correlated with the bait of co-IP described on the dataset and to the PRIDE project [21]. Proteins identified with a fold change up to 2, between the bait expression and the control of co-IP, are kept as potential interactors. The network draws on Cytoscape V.3.8.0 [23], the DyNet [24] application is used to compare the network publish in NDEx (according to [21]) and our result. A color code is given for nodes: red hexagon is the viral protein (bait), blue circles are the RefProts, and green circles are the AltProts, and for the edges: red means interaction not recovered in our analysis, grey means recovered in both analyses, green are specific to our analysis, and with a ratio <100 when purple edges are interaction specific to our analysis with a ratio of 100. A ratio of 100 means that protein is not detected in the control, and the expression can be link to the expression of the viral protein.

The AltProt identified (Appendix A) have been described based on the recovered information obtained from OpenProt database, Ensembl and RefSeq database.

Blast analysis (non-redundant sequences and RefSeq) of the AltProts sequences, identified in interaction with the SARS-CoV-2 proteins, show the presence of 27 AltProts exhibiting a homology rate greater than 80% (average of the coverage and identity percentage). These proteins, for a major part, are ncRNAs emitted, and are therefore not isoforms of homologous proteins because they originate from a different RNA sequence. From the total list of AltProts identified, Blast analysis revealed 16 AltProts with no significant (<80%) homologies; these 16 can have a known protein domain based on few identities with referenced protein, but experimental data are needed to proof the context of action to this AltProt. In the same way, 16 other AltProts have no Blast result in the human database (non-redundant sequences and RefSeq). In the context of following and understanding the SARS-CoV-2 way of action in the host cell, and considering the bat origin of the virus, the protein sequences of the no result blast were interrogated to the bats database (taxid:9397); 7 of the 16 AltProts describe similarity in bats protein, with a rate between 35% and 78% homology.

## 3. Results and Discussion

We studied the presence of potential AltProt involved in the interaction between the virus and the host cell, representing the possible role of the ghost proteome during a viral infection. The SARS-CoV-2 virus expresses a ~30 kb genome coding for at least 12 ORFs, able to produce at least 36 proteins (10 canonical + 26 nsps) [25,26] at the time of the study. Later research on the translational capabilities of viral RNA in host cells showed the presence of viral protein in reading frame shifts [27]; this could interestingly be considered as viral AltProt based on our previous definition of AltProt. The initial work [21] revealed the interaction between 332 human cellular RefProts with 27 viral proteins. Re-interrogation of these data using 23 viral targets, although some AltProt are known to be present at the level of the cell membrane, we focused our work on the viral proteins present in the cytoplasm, potentially involved in the replication mechanisms of the virus in the host cell. Including the AltProts database, this leads to identify 218 RefProts (common with the initial study), plus 56 AltProts involved in 93 interactions (Figure 1), of which 17 interacted with more than one viral protein. Moreover, 59% originate from ncRNA, 41% from mRNA, of which 39% were from the 3′UTR region, 34% from 5′UTR region, and 26% from a CDS shift (Table 1). Furthermore, 26 AltProts show identification only in the host cells (samples) for which the viral proteins have been expressed, and not in the control. These proteins are therefore specific for the stimulated condition, an expression variation cannot be determined, and so the sample/control ratio is equal to 100. The other 30, identified both under stimulation and in the control, are identified with a minimum of expression variation greater than or equal to two-fold changes. Some identified proteins and interactions are found to be different from the initial study because a different methodology was applied in the data reuse. This is a consequence of using a larger size database, including both RefProts and AltProts, then forcing the utilization of Proteome Discoverer in place of MaxQuant, following the recommendations of the OpenProt developers [18,28]. However, strong FDR filter is used, a unique peptide is verified for each identified protein, and a cutoff threshold sample/control of 2 is applied to define an interactor. Furthermore, 25 AltProts, after a Blast using a human nun-redundant database, present a strong homology (>80% of the average percentage of coverage and percentage of identity) to a RefProt, though they are identified with a unique peptide to the AltProt sequence. This case is not isoform because, coming from another gene of the RefProt, or from an ncRNA, share a common domain with the referenced or predicted protein. Global analysis of the biological processes of proteins identified as homologs shows that mainly the pathway impacted the protein metabolism (Figure 2A), in particular signaling pathways, such as protein translation and elongation (EIF2S2; EEF1A1; RPL35A; RPL4; RPS17; RPS18; RPL18A), and the regulation of protein synthesis by insulin (UBE2D3; HSPD1; HSPA8; PRKDC; HNRNPA1); interestingly proteins (RPL35A; RPL4; RPS17; RPS18; RPL18A) are found in the biological process of viral RNA translation, and in the pathway “Influenza Viral RNA Transcription and Replication”.

Interestingly, it was described that SARS-CoV-2 proteins impacted the phosphorylation state of the host cell proteins, such as the N protein, which was shown to differentially phosphorylate LARP1 and RRP9 [29]. In this way, it was not surprising to recover some AltProt with the riboprotein domain in interaction with SARS-CoV-2 proteins, such as IP_668819, IP_637436, IP_639311, IP_597129, IP_750273, and IP_667059. These proteins were identified as interacting with the non-structural proteins nsp8 (IP_637436, IP_750273, IP_667059) and nsp12 (IP_639311), two viral proteins described as being involved in the virus RNA replication [30,31,32]. Thus, finding interaction with the ribosomal protein and AltProt was not a surprise, in fact, the viral proteins nsp8 and nsp12 are described as interacting with the RNA of the host cell, at the same time, the ribosomal proteins are also fixed on the RNA, thus increasing their possibility of interaction. More than 37 ribosomal protein (RPL) can be observed in interaction with nsp8, RefProt, and AltProt confounding.

Historically the SARS Coronavirus (SARS-CoV) is known to be present in a large number of bats. Although the genome of these is less studied and annotated, genomic and proteomic data banks exist. Therefore, we observed if the AltProt sequences, with no homology with humans, could have some in bats. Of the 16 AltProts analyzed, 7 have a sequence homology, between 35% and 78%, with a bat protein. By their nature, unknown, and their unreferenced sequence, AltProts can present sequence similarities with other species, unexpected and not predicted until now. As a result, they could be the source of inter-species virus transmission, as well as the key to a new therapeutic approach in cases such as SARS-CoV-2 pathology.

The experiments carried out in this study make it possible to demonstrate the interactions of viral proteins with the proteins of the host cell. From this context, we have no information on the protein interactions inside of the host cell, so the determination of the functions of the identified AltProts is difficult, since the identified AltProts can be linked to all of the signaling pathways affected by the viral protein. Domain homology allows us to speculate on the function of these of the 27 AltProts with homology. For the others (32 proteins with <80% homology or without homology) considering their viral interacting protein and the RefProts that interact with these viral proteins, it is possible to hypothesize the signaling pathways involving these AltProts. In this way, among the five AltProts interacting with the viral protein “E”: IP_219869 (AltDGKH), IP_724315 (AltHMGN2P3), IP_788706 (AltEIF2S2P3), IP_555327 (AltAC006386.1) & IP_594707 (AltEEF1A1), three do not present an homology up to 80% with a RefProt (IP_219869, IP_555327, IP_594707); however, the study of Gene Ontology of RefProts found in interaction with E (Figure 2B), shows that the most represented Biological Processes are: “regulation of histone H3-K36 trimethylation” and “Synaptic vesicle budding from endosome” represented by the presence of RefProt: BRD4 and AP3B1. Thus, these three AltProt, such as IP_724315 (AltHMGN2P3) homologous to the “non-histone chromosomal protein HMG-17”, may be involved in modifications of histones or the chromosomal binding and, therefore, in epigenetic phenomena.

In the same way, six AltProts interact with the viral protein “M”, among them, two do not present any homology with RefProts. However, the other four are homologous with Tubulins family (TUBA3, TUBB2BP, and TUBAP2). Moreover, the Gene Ontology analysis of RefProts in interaction with M (Figure 2C) presents the main Biological Process: “microtubule nucleation by microtubule organizing center”. It is a safe bet that the two AltProts of unknown function are involved in microtubule organization and protein transport. Finally, the two AltProts (IP_671071, IP_565887), exhibiting low homology with bat proteins and observed in interaction with Orf8, can be proteins from the cytoskeleton, such as the AltProts IP_774695, IP_593099, IP_774693, and IP_656465, exhibiting strong homologies with the tubulin family, but may also be linked to the post-translational glycosylation modification signaling pathway, such as the Biological Processes of RefProts interacting with Orf8 (Figure 2D).

Overexpression of SARS-CoV-2 proteins in cell lines, followed by affinity purification and mass spectrometry of host proteins bound to the bait suggests an interaction, which need to be validated experimentally (i.e., “demonstrated”) using additional assays. Nevertheless, some AltProts are already foreseen to be key player in the virus-cell hijacking, such as AltHSPA8P11, which is found to interact with seven viral proteins. A cluster of AltProts centered on nsp6, nsp10, nsp11, Orf3b, Orf6, Orf7a, and Orf9b is also identified. Very interestingly, most of these proteins are involved in the interferon production inhibition, innate immunity modulation, cycle arrest, and host translation inhibition21. A major interest of the large scale interactomics is the possibility to screen for drug repurposing, as presented by the authors in their initial study. AltProts must now be considered as new potential therapeutic targets. Indeed, among the AltProts identified, the IP_2336782 (AltDUSP4) is found to be in interaction with Nsp6. AltDUSP4 shares 54% sequence homology with the C3a anaphylatoxin chemotactic receptor (C3AR1), which was recently shown to be involved in severe forms of COVID-19. C3AR1 is found over-activated in some patients, leading to a hyper-inflammatory profile, inducing persistence of the virus and a strong immunopathology [33]. Thus, AltDUSP4 is a potential target to reduce severe symptoms of COVID-19. Interestingly, the search for partner molecules via IUPHAR/BPS Guide to Pharmacology and BindingDB, shows the presence of sequence similarity between AltDUSP4 and the ATP binding cassette subfamily G member 2. It should be noted that the viral protein Nsp6 was previously identified as a target of Bafilomycin A1, a potent and selective inhibitor of the vacuolar H+-ATPase [21]. Several drugs are known to be active towards ATPase activity, e.g., cyclosporin A, KS 176, compound 14, Ko143, and Fumitremorgin C, and thus can target both NSP6 and AltDUSP4. 

Taken together, these new findings highlight the presence of many unknown proteins in the interactome between the host cells and the viral proteins that are involved in major pathways, such as innate immune response or translation regulation. Nevertheless, this study is a preliminary and descriptive study of AltProt identification in the previously published dataset, and requires dedicated research in order to specify the function and the role of these proteins in a strict way. This establishes that, besides the reference proteome, a ghost proteome exists, whose consideration would be highly beneficial both to the understanding of the pathophysiological mechanism of the virus and to establish therapeutic strategies.

## Authors Contribution

Conceptualization: M.S., I.F.; methodology: T.C.; formal analysis: T.C.; investigation: M.S., T.C.; resources: M.S., I.F.; data curation: T.C.; writing: I.F., M.S., T.C.; original draft: I.F., M.S., T.C.; supervision, project: I.F., M.S., T.C.; administration: M.S., I.F.; funding acquisition: I.F., M.S. All authors have read and agreed to the published version of the manuscript.

## Figures and Tables

**Figure 1 microorganisms-08-02036-f001:**
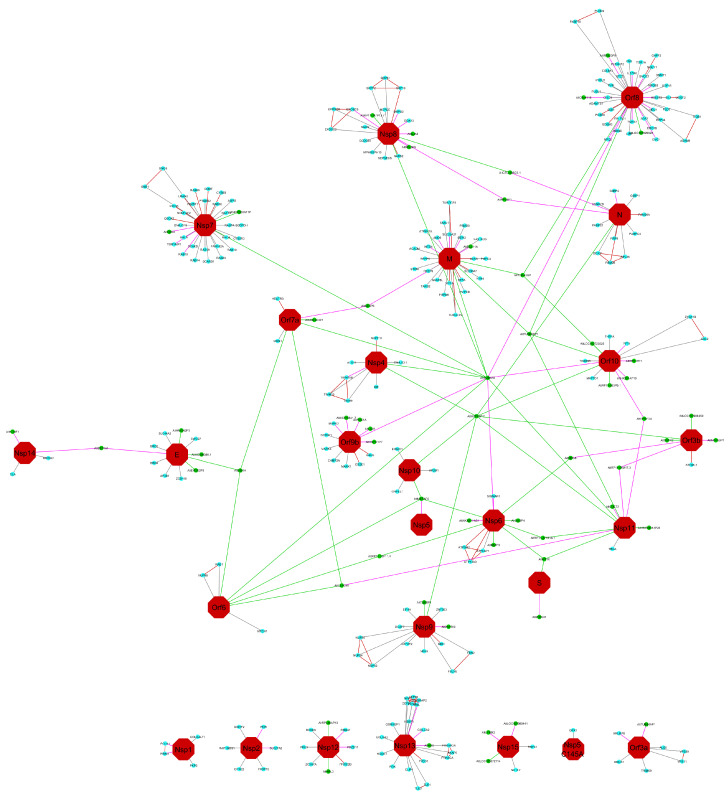
Viral protein interaction network (SARS-Cov-2) from co-IP experiments [21], re-analyzed by including AltProt, reference protein (RefProt), and viral databases. The previously established network is compared to the new query thanks to the DyNet Analyzer application on Cytoscape V3.8.0. Color legend nodes: red: viral protein (bait), blue: RefProts and green: AltProts, and for the edges: red: interaction not recovered in our analysis, grey are recovered in both analysis, green: specific to our analysis and with a ratio < 100, purple edges are interaction specific to our analysis with a ratio of 100.

**Figure 2 microorganisms-08-02036-f002:**
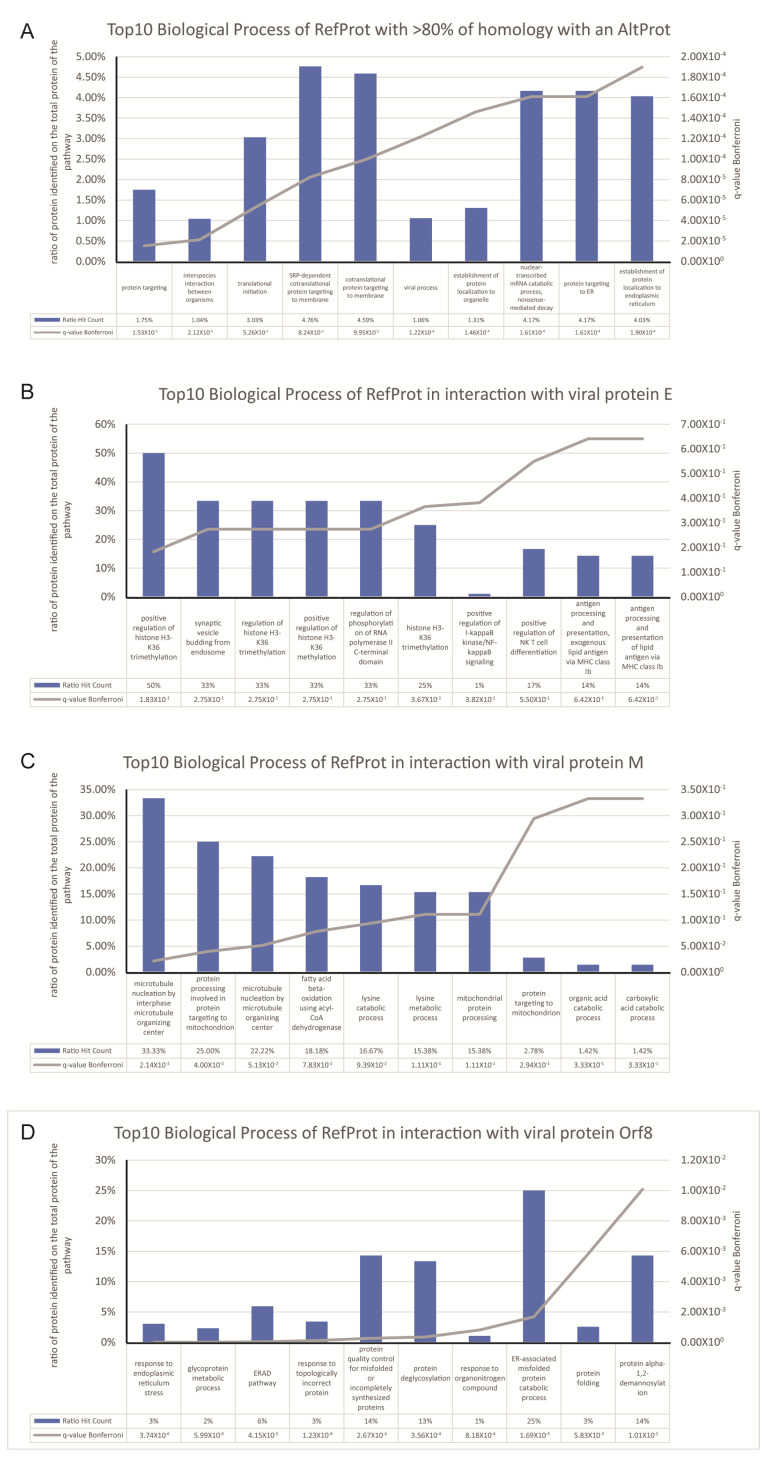
Gene Ontology analysis based on the RefProt identified in the network of interaction. ToppGene analysis is performed; (**A**) based on the gene name of the RefProt identified to have more than 80% of homology and on the RefProt identified in interaction with the bait of the co-IP; (**B**) RefProt in interaction with E; (**C**) in interaction with M; and (**D**) in interaction with Orf8. The pathway attributed to the RefProt is a clue for the AltProt link to the same bait.

**Table 1 microorganisms-08-02036-t001:** List of alternative proteins (AltProts) identified to be interacting with the SARS-Cov-2 viral proteins. The co-IP raw data were re-interrogated using OpenProt [18]. The table lists the 56 identified AltProts identified including the name of the gene coding for the RNA transcript, the accession number of the transcript, the name of the AltProt, the type of transcript for which AltProts are issued from and for AltProts originating from mRNA, the location on the mRNA.

Accession	GN	TA	AltProt	Type	Location	Viral Protein in Interaction
IP_555327	AC006386.1	ENST00000624491	AltAC006386.1	mRNA	3’UTR	E
IP_581922	AC018641.7	ENST00000435950	AltAC018641.7	ncRNA	-	orf9b
IP_659614	ANKRD20A11P	ENST00000442192	AltANKRD20A11P	ncRNA	-	nsp7
IP_187691	ARL3	NM_004311.3	AltARL3	mRNA	3’UTR	nsp12
IP_077449	BATF3	NM_018664.2	AltBATF3	mRNA	CDS	nsp6
IP_2387661	BCL11A	XM_017004337.1	AltBCL11A	mRNA	5’UTR	M
IP_565887	C9orf116	ENST00000371789	AltC9orf116	mRNA	5’UTR	orf8
IP_075271	CDC73	XM_006711537.3	AltCDC73	mRNA	CDS	nsp11
IP_766056	CEP290	ENST00000547691	AltCEP290	mRNA	5’UTR	nsp11 orf6 orf7a
IP_691726	CTC-398G3.1	ENST00000483614	AltCTC-398G3.1	ncRNA	-	N nsp8
IP_219869	DGKH	NM_152910.5	AltDGKH	mRNA	CDS	E orf6 orf7a
IP_2336782	DUSP4	XM_011544428.2	AltDUSP4	mRNA	5’UTR	nsp6
IP_235699	EDC3	ENST00000565602	AltEDC3	mRNA	3’UTR	nsp7
IP_594707	EEF1A1	ENST00000309268	AltEEF1A1	mRNA	5’UTR	E nsp14
IP_788706	EIF2S2P3	ENST00000428356	AltEIF2S2P3	ncRNA	-	E
IP_2396759	GJA5	XM_017001044.1	AltGJA5	mRNA	5’UTR	nsp8
IP_711582	HGS	ENST00000577012	AltHGS	mRNA	5’UTR	nsp13
IP_775502	HIGD1AP10	ENST00000527837	AltHIGD1AP10	ncRNA	-	orf10
IP_724315	HMGN2P3	ENST00000433603	AltHMGN2P3	ncRNA	-	E
IP_557348	HNRNPA1P28	ENST00000424481	AltHNRNPA1P28	ncRNA	-	nsp11
IP_572435	HSPA8P11	ENST00000508840	AltHSPA8P11	ncRNA	-	N nsp10 nsp11 nsp4 nsp9 orf10 orf3b
IP_658154	HSPD1P7	ENST00000447985	AltHSPD1P7	ncRNA	-	orf9b
IP_289249	KCNE1	XM_017028342.1	AltKCNE1	mRNA	3’UTR	nsp14
IP_075761	LAD1	ENST00000631576	AltLAD1	mRNA	CDS	orf9b
IP_671071	LOC101929023	ENST00000434879	AltLOC101929023	ncRNA	-	orf8
IP_2361135	LOC102723525	XR_925379.2	AltLOC102723525	ncRNA	-	orf10
IP_2268667	LOC105372714	XM_017028195.1	AltLOC105372714	mRNA	5’UTR	nsp15
IP_2266298	LOC107985441	XR_001754616.1	AltLOC107985441	ncRNA	-	nsp15
IP_2354489	LOC107986350	XR_001742414.1	AltLOC107986350	ncRNA	-	orf3b
IP_143572	LYRM2	NM_020466.4	AltLYRM2	mRNA	3’UTR	nsp15
IP_745252	MEG8	ENST00000553465	AltMEG8	ncRNA	-	nsp6 orf3b
IP_213668	METAP2	XM_005268583.3	AltMETAP2	mRNA	CDS	nsp10 nsp6 orf6
IP_729791	MT1X	ENST00000568370	AltMT1X	mRNA	3’UTR	nsp11 nsp6
IP_230046	NKX2-1-AS1	ENST00000521292	AltNKX2-1-AS1	ncRNA	-	nsp6
IP_597201	NOP56P1	ENST00000440030	AltNOP56P1	ncRNA	-	orf3b
IP_105102	POC1A	XM_011533561.1	AltPOC1A	mRNA	CDS	orf9b
IP_581419	RP11-10F11.4	ENST00000634439	AltRP11-10F11.4	ncRNA	-	nsp6 orf6
IP_734708	RP11-24M17.3	ENST00000567565	AltRP11-24M17.3	ncRNA	-	nsp11 orf3b
IP_667059	RP11-397P13.7	ENST00000427282	AltRP11-397P13.7	ncRNA	-	nsp8
IP_591742	RP11-471B18.1	ENST00000407538	AltRP11-471B18.1	ncRNA	-	nsp11 nsp6
IP_612631	RP11-553P9.1	ENST00000509116	AltRP11-553P9.1	ncRNA	-	orf10
IP_639311	RPL36AP13	ENST00000457490	AltRPL36AP13	ncRNA	-	nsp12
IP_750273	RPL4P1	ENST00000496596	AltRPL4P1	ncRNA	-	N nsp8
IP_637436	RPL5P9	ENST00000448118	AltRPL5P9	ncRNA	-	nsp8
IP_597129	RPS17P1	ENST00000396783	AltRPS17P1	ncRNA	-	orf10
IP_668819	RPS23P9	ENST00000448848	AltRPS23P9	ncRNA	-	orf8
IP_594653	SENP6	ENST00000474906	AltSENP6	ncRNA	-	M orf7a
IP_769089	SPRYD4	ENST00000338146	AltSPRYD4	mRNA	3’UTR	nsp11 orf10
IP_713094	SSTR2	ENST00000357585	AltSSTR2	mRNA	3’UTR	orf3b
IP_656465	TUBA3GP	ENST00000410028	AltTUBA3GP	ncRNA	-	M orf10 orf8
IP_774695	TUBAP2	ENST00000530835	AltTUBAP2	ncRNA	-	M nsp11 nsp4 nsp6 nsp7 nsp8 orf10 orf6 orf7a orf8 orf9b
IP_557241	TUBB4AP1	ENST00000450755.1	Alt TUBB4AP1	ncRNA	-	orf3a
IP_593099	TUBB2BP1	ENST00000404155	AltTUBB2BP1	ncRNA	-	M nsp11 orf10 orf8
IP_572422	TUBBP1	ENST00000518096	AltTUBBP1	ncRNA	-	nsp9
IP_665452	UBE2D3P1	ENST00000436669	AltUBE2D3P1	ncRNA	-	orf7a
IP_274314	ZNF569	XM_006723046.2	AltZNF569	mRNA	3’UTR	nsp9

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
