# Peer review of "SARS-Cov-2 Interactome with Human Ghost Proteome: A Neglected World Encompassing a Wealth of Biological Data"

_microorganisms, 2020, doi:10.3390/microorganisms8122036_

Round 1

Reviewer 1 Report

In this paper, the authors are describing the interactome map of SARS-CoV-2 with human cells, based on computational analysis. The authors have identified two very interesting points that worth mentioning 1) the use on online available tools to construct the virus-host interaction map and 2) identification of novel binding partners that can tailor the repurposing of pre-existing drugs that are available.

The authors should acknowledge the fact that their results depend of the experimental design of others. For example if any of the interactions described across different studies were done using either a different SARS-CoV-2 isolate, or virus concentration, or cell line etc you will automatically introduce bias in your analysis. Therefore I would suggest a more tempered tone through the manuscript and emphasize more their approach regarding computational analysis of binding partners.

Also I have a few minor comments:

line 53: please add non coding RNA

line 115: "but experimental data is need to proof the context of action to this AltProt" - this is a fair point and it should be mentioned through the manuscript. Especially regarding the use of this method and potential drawbacks 

line 122: please rephrase the first sentence; the tone should be more moderate since this is a prediction analysis

line 165: please change "protein" to "proteins"

Figure 1: please provide a high resolution image since it's impossible to see the interactions with the viral proteins

Figure 2: has the same problem as figure 1. I suggest that authors should use a pie chart to better show their analysis

discussion part: please provide, if known a list of potential drug candidates  (either presented as a separate table or Table 1). This will enhance the quality of the manuscript.

Author Response

Reviewer1:

In this paper, the authors are describing the interactome map of SARS-CoV-2 with human cells, based on computational analysis. The authors have identified two very interesting points that worth mentioning 1) the use on online available tools to construct the virus-host interaction map and 2) identification of novel binding partners that can tailor the repurposing of pre-existing drugs that are available.

The authors should acknowledge the fact that their results depend of the experimental design of others. For example if any of the interactions described across different studies were done using either a different SARS-CoV-2 isolate, or virus concentration, or cell line etc you will automatically introduce bias in your analysis. Therefore I would suggest a more tempered tone through the manuscript and emphasize more their approach regarding computational analysis of binding partners.

Also I have a few minor comments:

line 53: please add non coding RNA

According to the remark term “non-coding RNA” have been add in line 55-56 and removed in line 64.

line 115: "but experimental data is need to proof the context of action to this AltProt" - this is a fair point and it should be mentioned through the manuscript. Especially regarding the use of this method and potential drawbacks 

As a mention we add the sentence:” Nevertheless this study is a preliminary and descriptive study of AltProt identification in the previously published dataset and requires a dedicated research in order to specify the function and the role of these proteins in a strict way”

line 122: please rephrase the first sentence; the tone should be more moderate since this is a prediction analysis

in line 124, the first sentence have been change for “We studied the presence of potential AltProt involved in the interaction between the virus and the host cell, representing the role of the ghost proteome in this type of pathology.”

We hope in a more moderate tone that these AltProts exist, however the term "prediction" used by the reviewer is relative given the nature of the experiment. Certainly the database allowing the interrogation is carried out on prediction, as was a large number of RefProt. However, the analysis is based on the mass spectrometry data.

line 165: please change "protein" to "proteins"

line 168, “These protein are identified” have been changed by “these proteins are identified”

Figure 1: please provide a high resolution image since it's impossible to see the interactions with the viral proteins

Figure 2: has the same problem as figure 1. I suggest that authors should use a pie chart to better show their analysis

We are sorry for the poor quality of the images in the review manuscript, we are going to approach the editors in order to integrate the original images in 300PPI, .TIF

Figure 2, if we use a pie chart then some information will not be able to be represented.

discussion part: please provide, if known a list of potential drug candidates  (either presented as a separate table or Table 1). This will enhance the quality of the manuscript.

By their nature still very little explored, it is difficult to recommend potential drug candidates. The identification of these must be done on case-by-case, when information of the sequence presents a homology with known RefProt and targeted by a drug candidates, as we did for the example AltDUSP2. However on a large scale it also requires biological and physiological controls and verifications.

Reviewer 2 Report

In this manuscript, Cardon et al. use an existing dataset (Gordon et al, 2020) and interrogate a database containing Alternative ORFs to find potential Alternative Proteins predicted to interact with SARS-CoV-2 proteins. While this area of research is certainly relevant, also in the context of intracellular host-pathogen protein-protein interactions, the work presented here contains major concerns. 

The introduction of the work on which this analysis is based is very limited (lines 80-83) and would require additional details that are important to allow the reader to better interpret the findings. 

Lines 132-133: only 26 AltProts are specific to conditions where baits were expressed (Gordon et al,. 2020). Are the other "hits" non-specific (i.e. also found in the controls) yet still analyzed in this work?

It is not clear why the authors analysed the interactors of 23 (instead of 27) SARS-CoV-2 proteins and what criteria motivated this choice (lines 128-129).

Figures 1 and 2 are much to small. They are difficult to read and to interpret.

It would be interesting to add a column in table 1 to indicate the viral protein interacting with the AltProts. 

Key references missing (e.g. lines 229-230) or not up to date (e.g. line 127).

Vague and ambiguous statments (e.g. lines 16-17, 197, 199) or statments not supported by the data (e.g. lines 25-26, 179-180, 214-216, 221-222, 225-226).

Lines 173-178 are difficult to understand and the message provided by the authors is not clear. 

The conclusion that AltDUSP4 is involved in COVID-19 is highly speculative and could lead to misinterpretations by the reader. Additional experimental work formally demonstrating (or at least suggesting) it's involvment in SARS-CoV-2 biology is required. 

No statistics nor indicators of the likelyhood of predicted interactions are provided, making it difficult to assess the strenght of potential hits (canonical databases vs AltProts). 

Author Response

Reviewer2 :

In this manuscript, Cardon et al. use an existing dataset (Gordon et al, 2020) and interrogate a database containing Alternative ORFs to find potential Alternative Proteins predicted to interact with SARS-CoV-2 proteins. While this area of research is certainly relevant, also in the context of intracellular host-pathogen protein-protein interactions, the work presented here contains major concerns. 

The introduction of the work on which this analysis is based is very limited (lines 80-83) and would require additional details that are important to allow the reader to better interpret the findings. 

Additional description have been add to complete the introduction with a description of the method design.

Lines 132-133: only 26 AltProts are specific to conditions where baits were expressed (Gordon et al,. 2020). Are the other "hits" non-specific (i.e. also found in the controls) yet still analyzed in this work?

It is not clear why the authors analysed the interactors of 23 (instead of 27) SARS-CoV-2 proteins and what criteria motivated this choice (lines 128-129).

The CoIP technique used by the original authors of the SARS-CoV2 interaction article, is a technique known to be limited when identifying membrane proteins. Therefore we have chosen to exclude the viral proteins present in the membrane to concentrate on those present inside, which are also the most representative of the host cell's replication and modification mechanisms, while the membrane proteins are geared towards cell contamination.

Line 130-134, a sentence explaining our choice has been added: “although some AltProt are known to be present at the level of the cell membrane we have focused our work on the viral proteins present in the cytoplasm and potentially involved in the replication mechanisms of the virus in the host cell, moreover the CoIP method used for the identification of partners is known to have a limited efficiency on membrane proteins”

Figures 1 and 2 are much to small. They are difficult to read and to interpret.

We are sorry for the poor quality of the images in the review manuscript, we are going to approach the editors in order to integrate the original images in 300PPI, .TIF

It would be interesting to add a column in table 1 to indicate the viral protein interacting with the AltProts. 

As recommended by the reviewer one column have been add to the table:” Viral protein in interaction”

Key references missing (e.g. lines 229-230) or not up to date (e.g. line 127).

Some adjustment have been made on the reference cited in the article in way to be up to date

Vague and ambiguous statments (e.g. lines 16-17, 197, 199) or statments not supported by the data (e.g. lines 25-26, 179-180, 214-216, 221-222, 225-226).

Line 16-17 and 25-26 the abstract is established to draw the reader's attention to the main ideas of the article, moreover with a limited number of characters not all details can be included

Line 179-180 “The experiments carried out in this study make it possible to demonstrate the interactions of viral proteins with the proteins of the host cell” describes the context of the original study, the commentary here questions the original publication and the methodology used. In our context we have reprocessed the data previously obtained.

Line 197 “kind of tubulin” have been modified by “tubulin family (TUBA3, TUBB2BP, and TUBAP2)

Line 199 biological processes have established names

Line 214-216 “foreseen” is there to soften the tone of the sentence.

Line 221-222 “Indeed, among the AltProts identified, the IP_2336782 (AltDUSP4) is found to be in interaction with Nsp6” the sentence is supported by the data, this is an identification based on proteomic analysis, performed on mass spectrometry analysis. IP_2336782 is identified with a great score as the RefProt.

Line 225-226 “potential” is used to place a hypothesis, based on the results previously described: protein identification by mass spectrometry, protein data analysis, and sequence homology. Certainly the function, the role and the effectiveness as a therapeutic target remain to be proven in an experimental way, we pose here new hypotheses.

Lines 173-178 are difficult to understand and the message provided by the authors is not clear. 

This short paragraph have been change for: “Historically the SARS-CoV virus is known to be present in a large number of bats. Although the genome of these is less studied and annotated, genomic and proteomic data banks exist. Therefore, we have looked if the AltProt sequences with no homology with humans could have some in the bat. Of the 16 AltProts analyzed, 7 have a sequence homology, between 35 and 78%, with a bat protein. By their nature, unknown and their unreferenced sequence, AltProts can present sequence similarities with other species, unexpected and not predicted until now. As a result, they could be the source of inter-species contamination, as well as the key to new therapeutic approach in cases such as contamination with SARS-CoV-2.” In way to open the approach and the potential of the AltProt.

The conclusion that AltDUSP4 is involved in COVID-19 is highly speculative and could lead to misinterpretations by the reader. Additional experimental work formally demonstrating (or at least suggesting) it's involvment in SARS-CoV-2 biology is required. 

The overall tone of the paragraph describing AltDUSP4 is based on the hypothesis. The work presented in this article aims to open the field of possibilities in proteomics applied to a topical subject such as COVID. We have underlined the fact that additional experiments are necessary in order to use the AltProt described as a therapeutic target and / or to understand their function in the host cell or in interaction with the virus.

No statistics nor indicators of the likelyhood of predicted interactions are provided, making it difficult to assess the strenght of potential hits (canonical databases vs AltProts). 

Round 2

Reviewer 2 Report

In their revised manuscript, Cardon et al. have addressed some (but not all) points mentioned by the reviewer. Although, in their revised manuscript, the authors acknowledge that their analysis of an existing dataset is preliminary and that conclusions need to be addressed experimentally, some major concerns still remain. Additionally, the manuscript would greatly benefit from more explicitly relating the involvement of AltProts with processes assisting/restricting the intracellular replication cycle of SARS-CoV-2.

Specific comments:

  1. Lines 146-148: It is still not clear why the authors included or excluded proteins in their analysis. What do the authors refer to when mentioning "present at the level of the cell membrane"? The vast majority SARS-CoV-2 proteins is either expressed in the cytosol of the infected cell or is embedded in intracellular membranes. Notalbly, nsp4, nsp6, E and M (mentioned in table 1 and in the text) are "membrane proteins" yet included in the analysis. Please provide a clear list of SARS-CoV-2 proteins included in this analysis as well as inclusion/exclusion criteria.
  2. Line 153: "not found in the control". This suggests that 27/56 proteins are specific interactors of SARS-CoV-2 proteins. Are the other 29 proteins not specific? Please clarify.
  3. No statistics nor indicators of the likelihood of predicted interactions are provided, making it difficult to interpret the strength of potential interactors (coverage, number of peptides identified, …). A supplementary table could provide additional important information to interested readers.

Other comments:

  1. Lines 16-17, line 82: "involvement in development of the SARS-CoV-2 virus". This sentence is ambiguous, please clarify.
  2. Line 140: "this type of pathology". Please use more precise terms.
  3. Line 144: The number of SARS-CoV-2 ORFs and proteins is not accurate. ORF1a and 1ab encode for 16 non-structural proteins. The structural proteins S, E, M and N, as well as a set of accessory proteins (ORF3a, ORF6, ORF7a, ORF7b and ORF8, as well as potentially ORF3b, ORF9b and ORF10) are expressed by SARS-CoV-2.
  4. Line 152-153: "stimulated by the viral protein" is ambiguous. Please reformulate more precisely.
  5. Line 181: "and particularly in influenza" is ambiguous. Please reformulate more precisely.
  6. Line 189: "as the nsp8 and nsp12 complex on the RNA when the ribosomal protein work at the translation in protein" is ambiguous. Please reformulate more precisely.
  7. Line 192: do the authors refer the species Severe acute respiratory syndrome-related coronavirus, or to the SARS-CoV (or SARS-CoV-2) virus?
  8. Line 196-199: Please reformulate. It is not clear what "contaminations" refer to.
  9. Line 206: Overexpression of SARS-CoV-2 proteins in cell lines, followed by affinity purification and mass spectrometry of host proteins bound to the bait suggests an interaction, which needs to be validated experimentally (i.e. "demonstrated") using additional assays.

Author Response

In their revised manuscript, Cardon et al. have addressed some (but not all) points mentioned by the reviewer. Although, in their revised manuscript, the authors acknowledge that their analysis of an existing dataset is preliminary and that conclusions need to be addressed experimentally, some major concerns still remain. Additionally, the manuscript would greatly benefit from more explicitly relating the involvement of AltProts with processes assisting/restricting the intracellular replication cycle of SARS-CoV-2.

We thank the reviewer who paid attention to the work we provided. However, questioning the role of AltProt in the development of SARS-Cov-2, this is part of a full-fledged study which is not the purpose of this communication. Here we want to advise of the existence of these proteins and their possible involvement in interactions with viral proteins.

Specific comments:

  1. Lines 146-148: It is still not clear why the authors included or excluded proteins in their analysis. What do the authors refer to when mentioning "present at the level of the cell membrane"? The vast majority SARS-CoV-2 proteins is either expressed in the cytosol of the infected cell or is embedded in intracellular membranes. Notably, nsp4, nsp6, E and M (mentioned in table 1 and in the text) are "membrane proteins" yet included in the analysis. Please provide a clear list of SARS-CoV-2 proteins included in this analysis as well as inclusion/exclusion criteria.

Following the reviewer's comments, the viral proteins present in the Krogan study, but not considered in our study, were added. This addition induces a modification of the figure and of the associated identifications. Finally, only the ORF7b proteins, because described as being able to present too many false positive, and the ORF9c protein which is a prediction of viral protein have not been taken into consideration.

  1. Line 153: "not found in the control". This suggests that 27/56 proteins are specific interactors of SARS-CoV-2 proteins. Are the other 29 proteins not specific? Please clarify.

Yes, these 26 proteins are specific proteins of the stimulated condition, with no identification in the controls. For the 30 others, an identification in controls is obtained. However, we have kept only those which presented a variation of expression greater than two. This point is explained in line 113-114 “Protein identified with a fold change up to 2 between the bait expression and the control of CoIP are kept as potential interactor”.

The sentence have been changed and We add a sentence about the selection of the AltProt specifically variable:

“The other 30, identified both under stimulation and in the control, are identified with a minimum of expression variation greater than or equal to two fold changes.”

  1. No statistics nor indicators of the likelihood of predicted interactions are provided, making it difficult to interpret the strength of potential interactors (coverage, number of peptides identified). A supplementary table could provide additional important information to interested readers.

As requested by the reviewer, a table describing the score, abundance and ratio of AltProt identified for each viral protein and control sample have been added as a supplementary table 1

Other comments:

  1. Lines 16-17, line 82: "involvement in development of the SARS-CoV-2 virus". This sentence is ambiguous, please clarify.

Sentence have been change by : “Based on our experience with AltProts we have got interested in finding out their interaction with the viral protein coming from the SARS-CoV-2 virus, responsible for the 2020 Covid-19 outbreak”

  1. Line 140: "this type of pathology". Please use more precise terms.

Sentence have been changed by: “We studied the presence of potential AltProt involved in the interaction between the virus and the host cell, representing the possible role of the ghost proteome during a viral infection”

  1. Line 144: The number of SARS-CoV-2 ORFs and proteins is not accurate. ORF1a and 1ab encode for 16 non-structural proteins. The structural proteins S, E, M and N, as well as a set of accessory proteins (ORF3a, ORF6, ORF7a, ORF7b and ORF8, as well as potentially ORF3b, ORF9b and ORF10) are expressed by SARS-CoV-2.

It’s true that the number of proteins expressed by the virus has been updated and will probably increase as the team of Krogan already predicts from potential shift in the reading frame previously described. This underlines the parallel with our own purpose about the AltProt in host’s proteome. However, this number is based on the initial work of Krogan and has to be replaced on the context of the first study. Furthermore, this number explains the target used for the initial study and so the number is important and the sentence precise “at least” which calms the tone about this point.

  1. Line 152-153: "stimulated by the viral protein" is ambiguous. Please reformulate more precisely.

Sentence have been reformulated as: “26 AltProts show identification only in, the host cells (samples) for which the viral protein have been express and not in the control”

This sentence has been previously added because the reviewer needs more description about the term “sample”

  1. Line 181: "and particularly in influenza" is ambiguous. Please reformulate more precisely.

Sentence has been reformulated to contain the specific name of the super pathway: «Influenza Viral RNA Transcription and Replication”

(Based on: https://pathcards.genecards.org/card/influenza_viral_rna_transcription_and_replication)

  1. Line 189: "as the nsp8 and nsp12 complex on the RNA when the ribosomal protein work at the translation in protein" is ambiguous. Please reformulate more precisely.

Sentence have been reformulated: “So finding interaction with ribosomal protein and AltProt is not a surprised, in fact, the viral proteins nsp8 and nsp12 are described to interact with the RNA of the host cell, at the same time the ribosomal proteins are also fixed on the RNA, thus increasing their possibility of interaction.”

  1. Line 192: do the authors refer the species Severe acute respiratory syndrome-related coronavirus, or to the SARS-CoV (or SARS-CoV-2) virus?

Severe acute respiratory syndrome-related coronavirus is the “SARS-Cov” and referrers to the first strain of the SARS-CoV, sentence have been change trying to clarify this point:

“Historically the SARS Coronavirus (SARS-CoV) is known to be present”

  1. Line 196-199: Please reformulate. It is not clear what "contaminations" refer to.

“Contamination” have been replace: “As a result, they could be the source of inter-species virus transmission, as well as the key to new therapeutic approach in cases such as SARS-CoV-2 pathology”

  1. Line 206: Overexpression of SARS-CoV-2 proteins in cell lines, followed by affinity purification and mass spectrometry of host proteins bound to the bait suggests an interaction, which needs to be validated experimentally (i.e. "demonstrated") using additional assays.

The sentence suggested by the reviewer have been integrated to the manuscript in line 235-237 (numeration in simple mark).